# Tonsillectomy in Children with 22q11.2 Deletion Syndrome

**DOI:** 10.3390/genes13122187

**Published:** 2022-11-23

**Authors:** Jill M. Arganbright, Paul Bryan Hankey, Meghan Tracy, Srivats Narayanan, Janelle Noel-MacDonnell, David Ingram

**Affiliations:** 1Division of Otolaryngology, Children’s Mercy Hospital, 2401 Gillham Road, Kansas City, MO 64108, USA; 2School of Medicine, University of Missouri-Kansas City, 2411 Holmes St., Kansas City, MO 64108, USA; 3College of Osteopathic Medicine, Kansas City University, 1750 Independence Ave, Kansas City, MO 64106, USA; 4Department of Health Services and Outcomes Research, Children’s Mercy Hospital, 2401 Gillham Road, Kansas City, MO 64108, USA; 5Department of Sleep Medicine, Children’s Mercy Hospital, 2401 Gillham Road, Kansas City, MO 64108, USA

**Keywords:** 22q11.2 deletion syndrome, tonsillectomy, adenotonsillectomy, complications, obstructive sleep apnea

## Abstract

Tonsillectomy is one of the most common procedures performed in children, however there are currently no published studies evaluating tonsillectomy in children with 22q11.2 deletion syndrome (22q11DS). With this study, our goal was to investigate the indications, efficacy, and complications of tonsillectomy in a pediatric cohort of patients with 22q11DS. This is a retrospective chart review of patients in our 22q Center’s repository. Inclusion criteria were a diagnosis of 22q11DS and a history of tonsillectomy or adenotonsillectomy. Data collected included: indications for tonsillectomy, preoperative and postoperative polysomnography (PSG) results, and surgical complications. In total, 33 patients were included. Most common indications for tonsillectomy were facilitation with speech surgery (*n* = 21) and sleep-disordered breathing (SDB)/obstructive sleep apnea (OSA) (*n* = 16). Average length of stay was 1.15 days. Most patients (69%) had some degree of persistent OSA on postoperative PSG. Complications occurred in 18% of patients and included respiratory distress, hemorrhage, and hypocalcemia. This study demonstrates tonsillectomy was a commonly performed procedure in this cohort of patients with 22q11DS. These data highlight the potential need for close postoperative calcium and respiratory monitoring. The data were limited with respect to PSG outcomes, and future studies are needed to better characterize OSA outcomes and complications in this patient population.

## 1. Introduction

22q11.2 deletion syndrome (22q11DS) is the most common microdeletion syndrome, with an estimated prevalence of 1 in 3000 to 6000 children [1]. The variable clinical presentation most often includes congenital heart defects, palate anomalies, characteristic facial features, developmental delay, and/or intellectual disabilities, and immune dysfunction. Additional clinical involvement may include endocrine dysfunction, gastrointestinal anomalies, renal anomalies, hearing loss, and skeletal abnormalities [2].

Specific airway anomalies are also more prevalent in 22q11DS, including malacia, subglottic stenosis, retro/micrognathia, and midface hypoplasia [3,4]. These airway anomalies, as well as increased rates of hypotonia [5] and obesity [6], portend a higher risk of sleep-disordered breathing (SDB) and obstructive sleep apnea (OSA). Indeed, previous studies have demonstrated increased rates of SDB and OSA in children with 22q11DS compared to otherwise healthy children [7,8].

Tonsillectomy is a surgical procedure performed with or without adenoidectomy that completely removes the tonsils, including the capsule [9]. Tonsillectomy is one of the most common procedures performed on children, with more than 500,000 children undergoing the procedure annually in the United States [9]. The two most common indications for pediatric tonsillectomy are for treatment of SDB/OSA and recurrent tonsillitis [9]. Tonsillectomy and adenoidectomy (T/A) are often considered the first-line treatment for pediatric SDB/OSA, because these procedures are highly effective in non-syndromic patients [10]. However, it is much less clear whether tonsillectomy has similar outcomes in syndromic patients, with current studies often highlighting lower surgical cure rates for patients with Down syndrome [11] and Prader-Willi syndrome [12,13]. There are currently no published reports on outcomes for SDB/OSA after tonsillectomy in patients with 22q11DS.

Complications of tonsillectomy include postoperative respiratory problems and hemorrhage. The clinical practice guidelines for SDB/OSA treatment with tonsillectomy have identified certain populations at high risk for postoperative complications [9]. For these populations, preoperative sleep studies and postoperative overnight observation are recommended. High-risk populations broadly include patients with craniofacial anomalies, obesity, and neuromuscular disorders, all features associated with 22q11DS [2,9].

While it’s perceived tonsillectomy is a commonly pursued procedure to improve airway obstruction in patients with 22q11DS, there are currently no published studies evaluating tonsillectomy in this population. To that end, our goal was to investigate the indications, efficacy, and complications of tonsillectomy in a pediatric cohort of patients with 22q11DS.

## 2. Methods

### 2.1. Study Design and Participants

We reviewed the charts of patients enrolled in the repository for our tertiary care pediatric hospital’s 22q Center from September 2006 to May 2021. Inclusion criteria consisted of a diagnosis of 22q11DS and a history of tonsillectomy or adenotonsillectomy. Often patients undergo adenotonsillectomy, a procedure that removes the tonsils and adenoid tissue at the same time. When the tonsils alone are removed, the procedure is called tonsillectomy. To be as comprehensive as possible, we included patients who had undergone tonsillectomy alone as well as those who had undergone adenotonsillectomy. The specific procedure performed for each patient can be found in Table 1.

For each included patient, we collected genetic diagnosis, body mass index (BMI), past medical history including cardiac disease and palatal anomalies, indications for tonsillectomy, preoperative and postoperative polysomnography (PSG) results, length of hospital stay following tonsillectomy, postoperative calcium testing results, and any surgical complications. This study was approved by the Institutional Review Board at Children’s Mercy Hospital (IRB #00002008) with a waiver of consent.

### 2.2. Statistical Analysis

Data are descriptively summarized. Frequencies and percentages were used for categorical variables, and medians with range of values were provided for continuous variables. Comparisons were made between BMI at time of tonsillectomy and congenital heart disease (CHD) status, cleft palate anomalies, and tonsillectomy complications using Mann–Whitney U tests.

### 2.3. Sleep Study Results

Sleep studies were performed in a pediatric sleep laboratory accredited by the American Academy of Sleep Medicine (AASM). Patients were continuously monitored by a sleep technologist using a standard montage in accordance with AASM practice standards, including bilateral frontal, central, and occipital electroencephalogram (EEG), bilateral electrooculogram (EOG), chin electromyogram (EMG), bilateral anterior tibialis EMG, and single-lead electrocardiogram (ECG) for scoring sleep stage, and an evaluation of sleep architecture and limb movements. Respiratory parameters were monitored using oronasal thermal sensor and nasal pressure transducer for airflow, chest and abdominal inductive plethysmography for effort, body position, a snore sensor, and synchronized video monitoring. Arterial oxygen saturation was monitored using pulse oximetry. End tidal CO_2_ and/or transcutaneous monitoring was used to monitor carbon dioxide. Hypopneas were scored using a 3% desaturation or arousal rule. Sleep study results were reported by standardized reporting for pediatric sleep studies [14,15]. Specifically, we looked at the obstructive apnea-hypopnea index (OAHI) to operationalize the severity of OSA. The OAHI is the number of obstructive apneas and hypopneas that occur per hour of sleep. For pediatric patients, and for research purposes, an OAHI of <1/h is considered normal, an AHI of 1–5/h is mild OSA, an AHI of 5–10/h is moderate OSA, and an AHI > 10/h is severe OSA [14,15]. As the focus of the current investigation was on OSA in the context of tonsillectomy, the OAHI was the primary outcome of interest.

## 3. Results

### 3.1. Description of Study Participants

Of the 168 patients reviewed in the 22q repository, 141 had 22q11DS. Thirty-three (23%) of these patients had a documented tonsillectomy procedure at our institution and were included in the study. Fourteen (42%) of these patients were female and 19 (58%) were male. The majority of patients were White (*n* = 28), followed by Hispanic (*n* = 4), and multiracial (*n* = 1).

The mean age at tonsillectomy was 5.47 years (median 4.95, IQR: 3.91–6.41, SD 2.96). All patients (33/33) had a prior diagnosis of speech delay, 85% (28/33) had history of CHD, and 58% (19/33) had a history of palatal clefting (2 overt cleft palates, 17 submucous cleft palates). The mean BMI at the time of surgery was 15.77 kg/m^2^ (median 15.3, IQR:14.89–16.60, SD 1.38). There were no differences in the median BMI when comparing CHD (*p*-value = 0.5192) nor with cleft palate anomalies (*p*-value = 0.482). Twenty patients underwent T/A and 13 patients underwent tonsillectomy only (Table 1).

### 3.2. Indications for Tonsillectomy

We reviewed the documented indications for tonsillectomy listed as the preoperative diagnosis on the operative report. For some patients, multiple indications were documented, which were included in Table 1. The most common indication for tonsillectomy was the need for a first stage procedure for airway optimization prior to speech surgery (*n* = 21). Speech surgeries were defined as surgical procedures completed for the treatment of velopharyngeal dysfunction and in this study these procedures included pharyngeal flap (*n* = 14), sphincter pharyngoplasty (*n* = 3), and Furlow palatoplasty (*n* = 4). The second most common indication was management of SDB or OSA (*n* = 16). Less frequent indications included dysphagia (*n* = 3) and airway management preceding tracheostomy decannulation (*n* = 2).

### 3.3. Postoperative Course and Complications

The average length of hospital stay following tonsillectomy was 1.15 days after surgery (median 1.00, IQR: 1.00–1.00, SD 0.83). Most patients were scheduled as a planned overnight admission due to their complex medical history and/or consideration as “high risk” per the clinical practice guidelines for management of patients with SDB/OSA [9]. Within this cohort, 5 patients (15%) were discharged on the day of their tonsillectomy, 21 patients were admitted overnight (64%), 5 patients stayed for 2 days (15%), 1 patient for 3 days (3%), and 1 patient for 4 days (3%).

Complications occurred in 6/33 (18%) patients following tonsillectomy (Table 1) and included respiratory failure (*n* = 1), post-tonsillectomy hemorrhage (*n* = 3), and postoperative hypocalcemia (*n* = 4). Two patients had multiple complications. A detailed review of the reported complications was completed and assessed for any relationship with BMI at time of tonsillectomy. No notable relationship was found between complications and BMI (Mann–Whitney U *p*-value = 0.93).

#### 3.3.1. Respiratory Failure

One patient (Patient 10) experienced postoperative respiratory failure due to significant upper airway obstruction requiring overnight intubation in the pediatric intensive care unit (PICU). The patient was 1.56 years old at the time of the adenotonsillectomy. Closure of a tracheocutaneous fistula was completed concurrently. He was reported to have worsening obstructive breathing in the surgical recovery room and was transported to the PICU, where he was intubated. The patient was treated with intravenous (IV) steroids overnight and was extubated on postoperative day (POD) 1. He continued to require supplemental oxygen throughout the day, and IV steroids were continued for 24 h. Documented symptoms included considerable nasal congestion, which was felt to contribute to his obstructive breathing; oxymetazoline nasal spray twice daily was started. After a second night of monitoring in the PICU, the patient was able to be weaned from supplemental oxygen and was eventually discharged home on POD 2. Of note, this patient also had a complex medical history including right diaphragmatic paralysis and left vocal fold paresis. He had prior history of tracheostomy due to chronic lung disease and had been decannulated for 4 months prior to this surgery. It is unclear if the airway obstruction was related more to the adenotonsillectomy, the tracheocutaneous fistula closure, the patient’s other comorbidities, or a combination of these factors.

#### 3.3.2. Postoperative Hemorrhage

Three patients experienced reported post-tonsillectomy hemorrhage and sought care in an emergency department (ED).

Patient 6, aged 4.88 years, who underwent adenotonsillectomy, awoke with blood on his back, face, and chest on POD 7. The family presented to an external ED, where physicians did not find any active bleeding. After monitoring, it was determined he could be discharged home.

Patient 14, aged 9.95 years, experienced an episode of hemoptysis on POD 8 which was noted to have an active hemorrhage upon presentation to our institution’s ED. The patient was taken to the operating room for cauterization. A clot in the right tonsillar fossa was found and the surgery was uncomplicated. The patient was admitted for continued observation and an inpatient hematology consultation was obtained. Hematology recommended Amicar (aminocaproic acid) 3 times daily for 10 days. The patient’s recovery was otherwise unremarkable, and he was discharged home 3 days after cauterization. Of note, his outpatient hematology work-up 3 months after discharge showed mild chronic anemia but no concerns for bleeding disorder.

Patient 20, aged 3.91 years, had an episode of hemoptysis on the morning of POD 4 and was struggling with oral intake. The patient presented to our institution’s ED, where no active bleeding or clot were noted. To allow for close monitoring, the patient was admitted for observation and improved pain control. After 1 day of observation, no further bleeding was observed, oral intake had improved, and the patient was discharged home.

None of these 3 patients had a diagnosed bleeding disorder. One of the three patients had a Complete Blood Count (CBC) with differential within a year prior to surgery, and both platelet number and mean platelet volume (MPV) were normal; none of these patients had preoperative Platelet Function Assay (PFA). Traditionally, only post-tonsillectomy hemorrhage of patients who return to the operating room are factored into the calculation of postoperative hemorrhage rates. In our cohort, 1 out of 33 patients required surgical intervention for a post-tonsillectomy hemorrhage, resulting in an overall 3% postoperative hemorrhage rate.

#### 3.3.3. Postoperative Hypocalcemia

Hypocalcemia was defined as serum calcium < 8.6 mg/dL or ionized calcium < 1.2 mmol/L. Only 8/33 (24%) of patients received postoperative calcium monitoring, however of these, 50% (4/8) were found to have hypocalcemia. Patient 10 had a serum calcium of 7.8 mg/dL, Patient 19 had an ionized calcium of 0.76 mmol/L, Patient 20 had a serum calcium of 8.2 mg/dL, and Patient 23 had a serum calcium of 8.4 mg/dL. Each of these labs was drawn postoperatively within 24 h of the tonsillectomy. No patients had signs or symptoms of hypocalcemia. An inpatient endocrinology consult was obtained for two of the patients (Patient 19 and Patient 20). Patient 20 was treated with oral supplementation for 24 h, and Endocrinology recommended only repeat labs. Hypocalcemia was mild overall and did not have any known clinical sequalae. There was no indication that the patients’ hypocalcemia impacted their length of stay. Patients 19 and 23 were discharged home on POD 1 as anticipated, Patient 10 was discharged on POD 2 after spending 2 nights in the PICU for airway obstruction, and Patient 20 was discharged home on POD 2 as he was observed an extra day for poor pain control. All four patients with hypocalcemia eventually obtained normal calcium lab values: 2 had normalized calcium labs prior to discharge, 1 had normal calcium lab which was drawn on POD 10, and 1 had a normal calcium lab which was drawn 2 months postoperatively at an endocrinology visit. Preoperatively, Patients 10, 19, and 23 were noted to have normal serum calcium results within 6 months prior to surgery (Table 2). Patient 20 did not have calcium testing within 12 months of surgery, however they did have a normal calcium 2.45 years prior. Interestingly, Patients 10, 20, and 23 had a prior history of hypocalcemia, either shortly after birth or following a prior surgery (Table 2).

### 3.4. Sleep Study Results

We reviewed the results for patients who had sleep studies before and/or after tonsillectomy. We found 6 patients had preoperative sleep studies, 13 patients had postoperative sleep studies, and 5 patients had both preoperative and postoperative sleep studies (Table 1).

For the 6 patients who had sleep studies at some point prior to tonsillectomy, the average time from sleep study to tonsillectomy was 13.29 months (median 11.0, IQR: 7.27–16.78, SD 10.02). One patient’s sleep study was normal, 3 were diagnosed with mild OSA, 1 with moderate OSA, and 1 with severe OSA. The average obstructive AHI was 3.63 per hour (median 1.80, IQR: 1.40–4.30, SD 3.92).

For the 13 patients who had a sleep study at some point following tonsillectomy, 9 patients had some degree of persistent OSA and the remaining 4 were normal. Of these 9 patients who had findings of persistent OSA, 5 had mild OSA, 2 had moderate OSA, and 2 had severe OSA. The date ranges were also variable from the date of tonsillectomy to the time of sleep study. Sleep studies were performed an average of 46.13 months after surgery (median 26.37, IQR: 10.29–37.58, SD 55.44). The average obstructive AHI was 10.00 (median 2.80, IQR: 0.70–8.90, SD 17.55).

Five patients had both preoperative and postoperative sleep studies (Table 3). One patient did not have findings of OSA on either the preoperative or postoperative sleep study. One patient experienced resolution of their OSA, 1 had unchanged results, and the remaining 3 had persistent OSA on their postoperative sleep study. The date ranges of these studies are quite variable in relation to the date of the tonsillectomy procedure.

### 3.5. Platelet Testing Results

We reviewed CBC with differentials for the cohort to evaluate platelet count and MPV results in the twelve months prior to tonsillectomy. Results were recorded for 20/33 of patients. Of these, one patient had an abnormal platelet number of 100 (normal 150–450 10^3^/mcl). Four patients had an elevated MPV which shows larger than average platelet size. None of these patients with abnormal platelet testing experienced a post-tonsillectomy hemorrhage.

## 4. Discussion

In this study, we evaluated the indications, efficacy, and complications of tonsillectomy in a cohort of patients with 22q11DS. Interestingly, the most common indication for tonsillectomy was a staged procedure in preparation for speech surgery. Velopharyngeal dysfunction (VPD) is a common feature in patients with 22q11DS and often requires a speech surgery for treatment. Various types of surgeries can be done to help improve VPD, each of which itself causes some degree of airway obstruction. A staged tonsillectomy with or without adenoidectomy can be performed a few months prior to speech surgery. This procedure is often recommended to make more space in the airway prior to the speech surgery in the hope of limiting the risk of postoperative OSA following speech surgery [13]. Additionally, in some cases, tonsillectomy is needed to allow access to the surgical site(s) for the speech surgery [13]. In this study, airway optimization prior to speech surgery was indicated for 64% of patients. Similarly, tonsillectomy prior to speech surgery was also very common in a study by Lee et al., in which 77.5% of patients had tonsillectomy prior to wide pharyngeal flap to treat preoperative SDB/OSA and/or remove tonsils so as not to interfere with the pharyngeal flap surgery secondary to enlargement or positioning [13].

The second most common indication for tonsillectomy in this cohort was treatment of SDB/OSA, listed as a preoperative indication in 48% of our patients. Prior studies have shown OSA to be more common in children with 22q11DS. Kennedy et al. (2014) reviewed 323 patients with 22q11DS and found that 57 patients had completed at least one sleep study [7]. Of these, 33 had sleep study results showing OSA. These results demonstrated a prevalence of OSA in 10.2% (33/323) of patients, whereas the prevalence for the general pediatric population is close to 1–3% [7]. OSA within this cohort was most commonly mild in severity. The authors highlighted the importance of diagnosing and treating OSA in this population, as these patients often carry an underlying risk of cardiac and neuropsychiatric issues that can be exacerbated by OSA [7]. Another study by Silvestre et al. (2014) looked at the validated Pediatric Sleep Questionnaire (PSQ) results for 178 patients with syndromic cleft palate [8]. This study reported an overall 32% rate of positive screen for symptoms concerning for OSA; however, the subgroup with 22q11DS had the highest rate of positive scores (50%), which was a significant difference (*p* = 0.042). A study by Lee et al. (2020) looked at 40 patients with 22q11DS and VPD who were being evaluated for wide pharyngeal flap [13]. As part of the work-up, a preoperative sleep study was obtained, in which 37.5% of patients were diagnosed with OSA; most of these patients had mild OSA.

With adenotonsillectomy typically considered first line of treatment for OSA/SDB in non-syndromic children, we acknowledge the notable amount of literature that highlights less successful outcomes in syndromic patients, often attributed to multifactorial causes of OSA [11,16]. With the multitude of airway anomalies that are more common in children with 22q11DS [3,17], as well as hypotonia and obesity, OSA has the potential to be multifactorial and thus less likely to be successfully treated with tonsillectomy. We reviewed our sleep study data, particularly for patients with postoperative sleep studies, and found that it was common in our small sample to have persistent OSA following tonsillectomy. While our data suggest patients could still struggle with OSA even after tonsillectomy, our results were extremely limited with respect to sleep study outcomes.

Within our cohort, postoperative sleep studies occurred in only 13/33 patients. Postoperative sleep studies often occurred many months after the tonsillectomy, greatly increasing the potential for several significant confounding factors, including additional surgeries and health issues in the interim, which were not specifically investigated as part of this study. Additionally, there may be a selection bias affecting the number of patients with persistent OSA on postoperative sleep studies. Children whose symptoms of SDB/OSA were clinically resolved may have been less inclined to pursue a postoperative sleep study, and thus, their improvement/cure following the procedure may not be represented. Conversely, patients with persistent symptoms may be more inclined to undergo a postoperative sleep study, thus skewing our results toward showing persistent OSA. Given these possibilities, our current data do not lend themselves to conclusions regarding the outcomes of tonsillectomy for patients with 22q11DS in the treatment of OSA.

We do think these data highlight the need for additional studies, with a specific protocol to allow for timely preoperative and postoperative sleep studies for 22q11DS patients undergoing tonsillectomy for OSA/SDB. Ideally, sleep studies would be within 6 months before and 6 months after the procedure. We acknowledge that this type of a protocol can be challenging. Current hurdles we have encountered at our institution include the availability and timing of outpatient sleep studies, cost to families, and patients unable to tolerate the study. Future prospective studies are needed to better characterize OSA outcomes following tonsillectomy for patients with 22q11DS, particularly given the potential for these patients’ OSA to have multifactorial causes, resulting in potential less successful surgical outcomes compared to those without the syndrome.

We reviewed all tonsillectomy operative reports to determine the size of the patients’ tonsils. We were interested in tonsil size and how it relates to the treatment of OSA. With our small data set for patients with both pre- and postoperative sleep studies we again were not able to draw any formal conclusions. However, (Table 3) it is noteworthy that Patients 17, 31, and 33 all had very large tonsils and yet did not show any improvement on postoperative sleep study; this may suggest that OSA in patients with 22q11DS is multi-faceted and outcomes for OSA treatment may not be as predictable as in patients without the syndrome. More research is needed on this topic to further explore the relationship between tonsil size and OSA outcomes after tonsillectomy in this population.

From a complication standpoint, we had only one patient with respiratory compromise following tonsillectomy. Of note, this patient did have several risk factors for postoperative airway issues beyond 22q11DS; specifically, having a complex medical history that includes prior chronic lung disease, right diaphragmatic paralysis, and left vocal fold paresis, as well as concomitant closure of a tracheocutaneous fistula at the time of adenotonsillectomy. Given this complexity, multiple factors likely contributed to his airway distress following the surgery. Nonetheless, this case highlights the importance of close airway observation following tonsillectomy in patients with 22q11DS.

Post-tonsillectomy hemorrhage is a well-known complication of the tonsillectomy procedure. After tonsillectomy, the newly exposed wound bed, composed of muscle, heals over with keratinized squamous epithelium in a staged manger lasting 17 days. While the exact cause of postoperative hemorrhage is unknown, it’s been suggested that post-tonsillectomy hemorrhage occurs in the intermediate stage of healing when the fibrinous clot separates from the wound due to epithelial contracture from the surrounding ingrowth [18]. Other possible etiologies of postoperative hemorrhage could include the development and slough of a superficial eschar or a loosened vessel tie [19]. Often hemorrhage in these circumstances will clot soon after it begins and resolve with observation alone. If the patient is actively hemorrhaging upon presentation to the Emergency Department, they are typically taken to the operating room urgently for cauterization. This risk of hemorrhage after tonsillectomy that requires a second operative procedure for cauterization occurs in 3–4% of all tonsillectomy patients [20].

We were keenly interested in whether we would see an increased rate of postoperative hemorrhage in patients with 22q11DS following tonsillectomy, given the current literature discussing the potential risk for hematologic abnormalities in these children. Prior studies have shown that patients with 22q11DS can have abnormal platelet size, number, and function [21,22]. These platelet abnormalities are in part due to the location of the microdeletion on chromosome 22, which includes the GPIbβ gene. The GPIbβ gene encodes for one subunit of the platelet GPIb-V-IX receptor, which is critical for platelet adhesion [23,24]. For our entire cohort, 20 patients (61%) had a CBC with differential in the year leading up to surgery. Of these, 3 had elevated MPV, and 1 had both elevated MPV and low platelet number. As stated earlier, none of these patients had a post-tonsillectomy hemorrhage. However, given the risk for platelet abnormalities, surgeons could consider obtaining a CBC and Platelet Function Assay prior to performing tonsillectomy on children with 22q11DS. In our study, we had 1 patient who returned to the OR for re-cauterization of a post-tonsillectomy hemorrhage, resulting in a 3% hemorrhage rate for our cohort. This rate is consistent with the general population’s risk for post-tonsillectomy hemorrhage rate (3–4%) [20]. It is encouraging that this population of children with 22q11DS, despite risks of having potential hematologic abnormalities, experienced a similar post-tonsillectomy hemorrhage rate as the general population.

The third complication we encountered was postoperative hypocalcemia. Postoperative calcium monitoring was completed on only 24% of patients, however, of these, 50% showed hypocalcemia. Current management guidelines for children with 22q11DS [2] highlight the risk for hypocalcemia during times of biologic stress, which includes infection, burn, peripartum, and surgery. The guidelines recommend monitoring calcium during times of biologic stress [2]. It is hypothesized that the body’s increased demand for calcium due to the stress of surgery combined with a baseline diminished reserve of parathyroid hormone is the cause of postoperative hypocalcemia [25]. Patients undergoing this surgery may also have decreased oral intake both before and after tonsillectomy which may reduce their overall baseline oral intake of calcium. Although surgery is considered a type of biologic stress, and therefore a risk factor for hypocalcemia, little has been published on postoperative hypocalcemia and 22q11DS. The lead author’s current practice is to monitor patients with 22q11DS undergoing tonsillectomy overnight and check serum calcium on the morning of POD 1. Future studies are needed for insight into the clinical relevance of postoperative calcium monitoring following tonsillectomy for children with 22q11DS.

Three additional risks unique to patients with 22q11DS should be considered when performing adenotonsillectomy: causing or worsening VPD (particularly with adenoidectomy), carotid injury due to medialized vasculature, and cervical spine injury. Thankfully, we did not see any of these complications in our cohort, but these risks are important to discuss and consider when planning adenotonsillectomy on patients with 22q11DS.

Palate anomalies and subsequent velopharyngeal dysfunction are common in patients with 22q11DS [26]. VPD occurs when the palate cannot fully close against the back of the throat, allowing air to escape out of the nose while the child is speaking. The position of the adenoid pad is such that it can aid in the palate getting complete closure. Removing the adenoid tissue can, in some circumstances, create or worsen VPD [27,28]. Surgeons should carefully consider this risk when recommending adenotonsillectomy and should discuss the risk of VPD during the preoperative meeting. Surgeons can also consider performing only a tonsillectomy without performing an adenoidectomy or can consider removing only a small portion of the adenoid tissue (superior adenoidectomy) with the hope of limiting the risk of creating VPD. The risks and benefits for removing all, part, or none of the adenoid tissue at the time of tonsillectomy needs to be personalized to the individual patient, and surgeons should carefully discuss recommendations and risks with families.

Vascular anomalies are very common in patients with 22q11DS. A study by Oppenheimer et al. found that 93% of patients with 22q11DS had some form of cervical vascular anomaly. Specifically, a medial deviation of the carotid artery position has been reported in 30–49% of patients [29]. Some studies have recommended obtaining imaging to evaluate for medialized vasculature prior to performing any oropharyngeal procedure (including tonsillectomy) [30]. The utility and need for preoperative imaging is currently controversial [31,32], with studies discussing the cost-effectiveness as well as the potential risks of imaging to patients (e.g., MR angiogram often requires sedation, and CT angiogram involves exposure to radiation). Currently, no studies address risks of medialized vasculature specifically as it relates to tonsillectomy. Regardless of the surgeon’s current protocol for preoperative imaging, it is important to consider this risk when performing tonsillectomy on patients with 22q11DS and, at the least, carefully inspect and palpate the tonsillar fossa prior to incision and throughout the case.

Lastly, congenital cervical anomalies are extremely common in children with 22q11DS [33,34]. A recent systematic review found at least one cervical or occipital anomaly in over 90% of cases. This risk of cervical anomalies is important to be aware of, particularly as tonsillectomy surgery requires the patient to be positioned with their neck extended. The focus while positioning the patient should be on using the least amount of extension needed to perform the procedure safely. The practice guidelines for management of 22q11DS recommend obtaining flexion and extension c-spine XR films around age 4 years or when the patient is able to participate [2]. If these films are available to the surgeon, they should be reviewed prior to tonsillectomy to ensure no known cervical spine abnormalities exist.

Limitations to this study include its retrospective nature and small sample size. The limited patient number, extremely variable time between preoperative and postoperative sleep studies, as well as the high risk for selection bias, did not allow us to make any conclusive statements regarding the outcomes of tonsillectomy for treatment of OSA/SDB. Additionally, with the limited number of patients who had pre- and postoperative sleep studies, we were unable to fully assess the relationship of tonsil size to OSA outcomes following tonsillectomy. Our data on the complication of postoperative hypocalcemia is limited; we have preoperative calcium within 6 months of the surgery for three of the four patients with reported hypocalcemia which were normal, and the fourth patient had a serum calcium that was normal 2.45 years prior to the surgery. As we do not have calcium levels within 24–48 h before the surgery, it does make it hard to determine that the hypocalcemia noted after surgery was not there prior to the surgery, in which case it would be a pre-existing condition and not a complication of the surgery. From a complication standpoint, this study is limited to those complications that either presented to our facility or that were documented in the electronic medical record. If a complication (i.e., post-tonsillectomy hemorrhage) occurred and patients presented to outside facility, it is possible we were not aware of this complication and it was thus not included in the study.

## 5. Conclusions

Tonsillectomy was a commonly performed procedure in this cohort of patients with 22q11DS, with facilitation of speech surgery and treatment of SDB/OSA as the most common indications. Most patients spent one night in the hospital prior to discharge. Our data were limited with respect to sleep study outcomes, where future studies are still needed to better understand OSA outcomes status post tonsillectomy for children with 22q11DS. Complications included respiratory distress and postoperative hypocalcemia. The postoperative hemorrhage rate was similar to the general population following tonsillectomy. These data highlight the need for larger prospective studies to better understand the outcomes of tonsillectomy and to better characterize associated complications.

## Figures and Tables

**Table 1 genes-13-02187-t001:** Data Summary for Patients with 22q11DS and Tonsillectomy. The shaded area contains all indications for tonsillectomy for each patient.

						Indications for Tonsillectomy						
Patient #	CHD	Cleft Palate Anomaly	Age at Surgery (Years)	Surgery	Tonsil Size (+) *	VPD	Dysphagia	SDB/ OSA	Airway Management Prior to Decann	PSG Before Surgery	PSG After Surgery	Complications
1	+	+ ^~^	16.77	T	3	+				−	−	
2	+	+ ^~^	4.12	T	Unk	+				−	−	
3	+	−	3.71	T	3		+			−	+	
4	+	−	8.45	T	2	+				−	−	
5	+	+ ^~^	5.98	T	2	+				+	+	
6	+	−	4.88	T/A	3	+				−	−	b ^1^
7	+	−	6.85	T	2				+	+	+	
8	−	+ ^~^	4.95	T/A	2–3			+		−	−	
9	−	+ ^~^	4.08	T	1	+				−	+	
10	+	−	1.56	T/A	3			+		−	−	a, c
11	+	−	6.41	T/A	2	+				−	−	
12	−	−	4.50	T/A	3–4			+		−	−	
13	+	+ ^~^	4.61	T	4	+		+		−	−	
14	+	+ ^~^	9.95	T/A	3	+		+		−	+	b ^2^
15	+	+ ^~^	6.53	T	4	+		+		+	−	
16	+	+ ^~^	9.72	T/A	3	+		+		−	−	
17	+	+ ^~^	1.89	T	3–4			+		+	+	
18	+	−	8.72	T	3	+				−	−	
19	+	+ ^~^	5.38	T/A	3	+				−	−	c
20	+	+ ^~^	3.91	T/A	3	+				−	+	b ^1^, c
21	+	+ ^~^	5.95	T/A	3			+		−	−	
22	+	−	4.69	T/A	3	+		+		−	+	
23	+	+^~^	5.33	T/A	1–2	+		+		−	+	c
24	+	+ ^~^	5.28	T/A	3	+				−	−	
25	−	−	2.58	T/A	3–4		+			−	−	
26	+	−	2.61	T	2				+	−	+	
27	+	+ ^~^	5.39	T/A	2	+				−	+	
28	+	−	8.16	T/A	2	+		+		−	−	
29	+	+ ^~^	1.62	T	4		+	+		−	−	
30	+	+ ^&^	5.33	T/A	2	+				−	−	
31	+	+ ^&^	3.11	T/A	4			+		+	+	
32	−	−	4.18	T/A	3	+		+		−	−	
33	+	−	3.43	T/A	3			+		+	+	

CHD, congenital heart disease; ^~^ submucous cleft palate; ^&^ overt cleft palate; T, tonsillectomy; T/A, adenotonsillectomy; * Tonsil size as reported in tonsillectomy operative note; VPD, velopharyngeal dysfunction; SDB, sleep disordered breathing; OSA, obstructive sleep apnea; Decann, decannulation; PSG, polysomnogram (sleep study); a = airway obstruction; b = post-tonsillectomy hemorrhage (1 = observation, 2 = Return to OR); c = hypocalcemia.

**Table 2 genes-13-02187-t002:** Patients with postoperative hypocalcemia.

Patient #	History of Hypocalcemia	Preoperative Calcium	Time from Result to Surgery (Days)	Postoperative Calcium
10	Transient in NICU	10.2 mg/dL ^1^	174	7.8 mg/dL ^1^
19	No history	9.6 mg/dL ^1^	153	0.76 mmol/L ^2^
20	Transient in NICU and during childhood	9.7 mg/dL ^1^	895	8.2 mg/dL ^1^
23	Transient following previous surgery	9.3 mg/dL ^1^	111	8.4 mg/dL ^1^

^1^ serum calcium (hypocalcemia ≤ 8.6 mg/dL); ^2^ ionized calcium (hypocalcemia ≤ 1.2 mmol/L).

**Table 3 genes-13-02187-t003:** Preoperative and postoperative polysomnography. Polysomnography results are highlighted in gray areas.

	Preoperative PSG						Postoperative PSG			
Patient #	BMI (kg/m^2^)	Time From PSG to Surg (Months)	Obstructive AHI	OSA Severity	Surgery Performed	Tonsil Size	BMI (kg/m^2^)	Time From Surg to PSG (Months)	Obstructive AHI	OSA Severity
5	16.7	11.28	0.75	normal	T	2	23.7	138.58	0.2	normal
7	14.87	18.61	1.3	mild	T	2	21.47	67.3	0	normal
17	16.1	6.12	1.9	mild	T	3–4	Unk *	32.32	2.8	mild
31	16.36	10.72	5.1	moderate	T/A	4	14.6	35.34	49	severe
33	18.55	2.53	11	severe	T/A	3	17.09	10.13	48.89	severe

* Unk = Unknown.

## Data Availability

The data presented in this study are available on request from the corresponding author. The data are not publicly available due to privacy.

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
