# Peer review of "Tonsillectomy in Children with 22q11.2 Deletion Syndrome"

_genes, 2022, doi:10.3390/genes13122187_

Round 1

Reviewer 1 Report

Dear authors,

This is an interesting article regarding tonsillectomy in children with a genetic problem. I have some questions and suggestions.

Abstract

- line 24 - please define OSA, even in Abstract

Methods

Line 100-101 - please define EEG, EOG, EMG, ECG

Line 77, 115, 240, 390 - please check the numbering

Results

Line 153 and 198- How do you define  "postoperative hypocalcemia"? Were calcium levels measured preoperatively? If not, can hypocalcemia be defined as a complication or was pre-existing? 

 Line 161 - please define IV 

For the three patients with postoperative hemorrhage a few days after the surgical intervention, do you have any explanation why this complication occurred, without the patients presenting coagulation abnormalities?

In your cohort you studied body mass index, past medical history including cardiac disease and palatal anomalies. I would suggest checking if there is any correlation between these factors and postoperative complications.

Discussion

Line 315-343 - you discuss about  abnormal platelet size, number, and function. Did you perform these studies on the analyzed patients? Same for calcium levels, pre and postoperatively. 

More limitation of the study should be added.

Reviewer 2 Report

This article is well written study about tonsillectomy patients with 22q deletion syndrome. 
As you wrote in the article, I was sorry for the OSA results which did not agree with previous studies. 

I wonder that is there any indication for performing PSG preoperative or postoperatively? At the table I, there were many patients who suffered from SDB or OSA, but only 13 patients performed PSG. 

As you wrote in the discussion part, I wonder the adenoidectomy can deteriorate the VPI status. 

Is there any information about tonsil size? According to tonsil size, it can be helpful to reduce the airway problems or not.

Round 2

Reviewer 1 Report

Dear Authors,

I agree that the manuscript has been improved. However, I would like you to mention in the abstract that the calcium level was not measured preoperatively.

Regards. 

Reviewer 2 Report

The authors tried to revised the manuscrips as pointed well.

I think this article is well written study which included patients with genetic disease 22q11DS.

However, I think this kind of study is not fit in this journal "Genes". It might be appropirate in pediatric or otorhinolaryngology journals which presented many clinical studies.